# Cysteine-rich intestinal protein 1 is a novel surface marker for human myometrial stem/progenitor cells

Emmanuel N. Paul [1], Tyler J. Carpenter[1], Sarah Fitch[1,2], Rachael Sheridan[3], Kin H. Lau [4], Ripla Arora [1,2] & Jose M. Teixeira [1✉]

Myometrial stem/progenitor cells (MyoSPCs) have been proposed as the cells of origin for uterine fibroids, but the identity of the MyoSPC has not been well established. We previously identified SUSD2 as a possible MyoSPC marker, but the relatively poor enrichment in stem cell characteristics of SUSD2+ over SUSD2- cells compelled us to find better markers. We combined bulk RNA-seq of SUSD2+/- cells with single cell RNA-seq to identify markers for MyoSPCs. We observed seven distinct cell clusters within the myometrium, with the vascular myocyte cluster most highly enriched for MyoSPC characteristics and markers. *CRIP1* expression was found highly upregulated by both techniques and was used as a marker to sort CRIP1+/PECAM1- cells that were both enriched for colony forming potential and able to differentiate into mesenchymal lineages, suggesting that CRIP1+/PECAM1- cells could be used to better study the etiology of uterine fibroids.

[1] Department of Obstetrics, Gynecology and Reproductive Biology, College of Human Medicine, Michigan State University, Grand Rapids, MI 49503, USA. [2] Institute for Quantitative Health Science and Engineering, East Lansing, MI 48824, USA. [3] Flow Cytometry Core, Van Andel Institute, Grand Rapids, MI 49503, USA. [4] Bioinformatics and Biostatistics Core, Van Andel Institute, Grand Rapids, MI 49503, USA. ✉email: teixei15@msu.edu

Uterine fibroids, also known as leiomyomas, are benign tumors found in the smooth muscle layer of the uterus, the myometrium. Uterine fibroids develop in up to 80% of women during their reproductive years and, although benign, are often associated with debilitating symptoms such as menorrhagia, anemia, dysmenorrhea, pelvic pain, and urinary incontinence[1,2]. Hormonal therapies, mainly used for alleviating fibroid symptoms, are generally short-term treatments due to long-term side effects or induced infertility[3]. Hysterectomy, the most common and effective treatment for uterine fibroids, results in permanent infertility[4]. Progress in the search for effective medical therapies that preserve fertility and avoid invasive surgery has been difficult, in part because fibroid etiology and pathogenesis of the disease is unclear.

A dysregulated myometrial stem/progenitor cell (MyoSPC) has been proposed as the cell of origin for uterine fibroids. After embryonic development, tissue-specific stem cells remain throughout the body and play important roles in tissue homeostasis, including replacing dying cells and participating in tissue remodeling[5]. The dramatic remodeling that occurs during pregnancy and following parturition in the uterus suggests a need for and the existence of myometrial stem cells[6]. Uterine fibroids are thought to be a clonal disease[7–9], and since most clonal diseases have a single cell origin[10], we and others[11,12] have hypothesized that a mutated MyoSPC could be the cell of origin for uterine fibroids[13]. Thus, the identification of the MyoSPC has been an important goal of many laboratories to begin studying the underlying mechanisms of fibroid etiology. The presence of cells with stem cell properties has been demonstrated using the label-retaining cells in a mouse model[14,15] and using the side population (SP) method in human myometrium[16,17]. Putative MyoSPCs have been isolated and studied by using a combination of cell surface markers, including SUSD2[18], CD44/Stro-1[11] and CD34/CD49f/b[19]. However, cell surface markers in these studies have been selected using stem cell markers from other tissues, and their respective contributions to myometrial smooth muscle regeneration have not been well established. We and others have also used the side population (SP) discrimination assay[16,17], but this stem cell identification technique has multiple pitfalls, not the least of which is difficulty in enriching and recovering live SP cells for further analyses[20]. Because the endometrial stroma and the myometrium originate from the same embryonic tissue, the Müllerian duct mesenchyme[6], we recently proposed that SUSD2, an endometrium stem cell marker[21], also enriches for MyoSPCs[18]. While SUSD2+ cells do have mesenchymal stem cell characteristics, SUSD2+ cells represent between 25-40% of total myometrial cells. Additionally, colony formation is only increased 2.8-fold increase in SUSD2+ cells compared to the rest of the myometrial cells, suggesting that further enrichment might be possible. The objective of the present study was to integrate next-generation sequencing, including single cell RNA-seq and bulk RNA-seq, to identify a more specific marker to significantly enrich for MyoSPCs from human myometrium, which can then be used to better understand the molecular mechanisms underlying fibroid etiology. In the present study, we found that CRIP1 expression was highly upregulated by both sequencing techniques and could be used to enrich MyoSPCs further.

## Results

**SUSD2+ are enriched for characteristic MSC genes compared with SUSD2− cells.** To determine how best to enrich for stem cell activities in the SUSD2+ MyoSPC population, we used SUSD2 to enrich for myometrium stem cells followed by RNA-seq to discover new cell surface markers for MyoSPCs in the human myometrium. Myometrial cells from non-fibroid patients (MyoN, $n = 5$) were isolated and live SUSD2+ and SUSD2- cells were

sorted by flow cytometry. As with our previous results[18], 30–50% of the myometrial cells were SUSD2+ (Fig. 1a). Total RNA was isolated from the two cell populations and sequenced for differential gene expression analyses. Principal Component Analysis (PCA) plot showed that SUSD2+ and SUSD2- cells were separated by principal component 1 with a variance of 39%, indicating a strong divergence in the transcriptomic profiles of these two cell populations (Fig. 1b). A total of 6777 significant differentially expressed genes (DEGs) were detected between SUSD2+ and SUSD2- myometrial cells with a p-adjusted false discovery rate (FDR) < 0.05 (Fig. 1c and Supplementary Data 1). 3527 genes were down-regulated and 3250 were upregulated in the SUSD2+ population compared to the SUSD2- population. We confirmed that SUSD2 was upregulated in the SUSD2+ sorted cells and that they were also enriched in other MSC markers such as MCAM, PDGFRβ and CSPG4 (Fig. 1c, d). A heatmap of the top 300 DEGs from the SUSD2+ to SUSD2- cells comparison showed separation of the cell types and included SUSD2, MCAM, PDGFRβ and CSPG4 as differentially expressed genes (Fig. 1e).

**Myometrium side population cells are not enriched in MSC markers.** The side population (SP) phenotype is another often used method to isolate cells with stem cell characteristics that exploits the ability of some stem cells to efflux the DNA-binding dye Hoechst 33342 via the ATP-binding cassette (ABC) transporters[11,16,17,22,23]. An average of 1.7% of the total MyoN myometrial cells from 3 different patients were SP+ (Fig. 2a). Addition of verapamil, a calcium channel blocker used as a negative control to validate the SP, severely decreased the number of the myometrium SP+ cells (Fig. 2b). SP+ and SP- myometrial cells were sorted for total RNA sequencing and analyzed by PCA plot, which showed that matched SP+ and SP- cells segregated by the principal component 2, accounting for 25% of the variance (Fig. 2c). A total of 828 significant (FDR < 0.05) DEGs, including 478 upregulated genes and 350 downregulated genes, were detected between the SP+ and SP- myometrial cells (Supplementary Data 2). The top 10 DEGs enriched in the SP+ to SP- comparison were associated with immune response (XCL2[24], CD69[25], IL7R[26], KLRD1[27], and IL18R1[28]) apoptosis (TNFRSF10A[29]), extracellular matrix (SPOCK2[30]) and hematopoietic stem cells (SELE[31], GATA3[32], CD69[33], and VCAM1[34]) (Fig. 2d). We confirmed an increase in expression of two major ABC transporters, including ABCB1, and ABCG2, and a decrease in PGR, another marker previously shown downregulated in the SP+ compared to the SP- of human myometrial cells[16] (Fig. 2e). Surprisingly, SP+ cells did not show increased expression of putative MSC markers[19,35,36], SUSD2, MCAM, PDGFRβ, CSPG4, CD44, CD34 and ITGA6 (also known as CD49f) compared to the SP- (Fig. 2f).

**A putative MyoSPC cluster is determined by single cell RNA-seq.** A total of 9,775 cells from MyoN myometrium samples from 5 different patients passed quality control with an average of 98.3% sequencing saturation, or approximately 512,000 reads per cell. Uniform Manifold Approximation and Projection (UMAP) of myometrial ($n = 5$) single cell RNA-seq (scRNA-seq) revealed 7 main cell clusters (Fig. 3a) with similar cell distribution patterns across the five samples (Supplementary Fig. 1a). Cluster identities were assigned using the expression profiles of canonical markers for cell populations expected to be found in the myometrium (Fig. 3b)[37,38], including 4 different smooth muscle cell types, vascular myocytes, myocytes, myofibroblasts, and fibroblasts. The cell proportion of each identified cluster was similar across patients with these muscle cell types dominant (Supplementary Fig. 1b). Four MSC markers, SUSD2, MCAM, PDGFRβ and CSPG4, were found highly expressed in the vascular myocyte

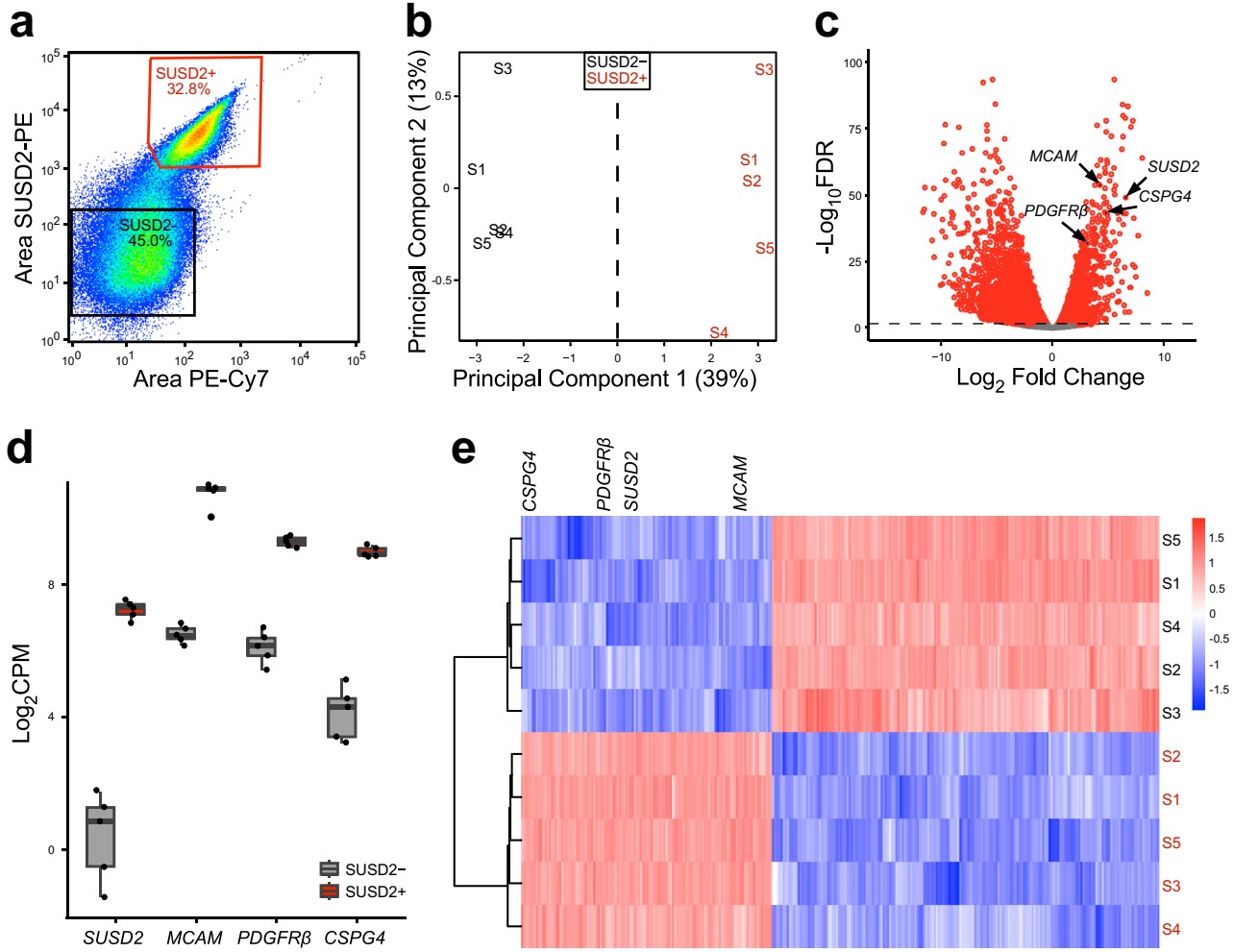

**Fig. 1 Bulk RNA-seq results of SUSD2- and SUSD2+ cells from myometrial samples. a** Representative flow cytometry scatter plot of single cell SUSD2-
and SUSD2+ sort showing over 30% of the live myometrial cells positive for the SUSD2 marker. Boxed areas indicate gating strategy. **b** Principal
component analysis (PCA) plot of RNA-seq results from SUSD2- (in black) and SUSD2+ (in red) cells, labels represent individual patient samples ($n = 5$),
and variance for each PC is indicated in percentage. **c** Volcano plot showing up ($n = 3527$) and down ($n = 3250$) DEGs with an FDR-adjusted $p$-
value < 0.05 in SUSD2+ vs. SUSD2- cells depicted as red dots. Gray dots represent genes with an FDR $p$-value > 0.05. **d** Boxplot of mesenchymal stem
cells markers, *SUSD2*, *MCAM*, *PDGFRβ* and *CSPG4* in the SUSD2- (in gray) and SUSD2+ (in red) cell population ($n = 5$). Gene expression is shown as
log$_2$CPM. SUSD2+ sorted cells are significantly enriched for MSC markers including *SUSD2* (log$_2$FC = 4.5, FDR $p = 2.9 \times 10^{-5}$), *MCAM* (log$_2$FC = 3.1, FDR
$p = 8.8 \times 10^{-16}$), *PDGFRβ* (log$_2$FC = 2.8, FDR $p = 1.5 \times 10^{-11}$) and *CSPG4* (log$_2$FC = 3.4, FDR $p = 1.1 \times 10^{-11}$). **e** Heatmap of the top 300 DEGs from SUSD2+
vs. SUSD2- cells comparison with unsupervised hierarchical clustering of genes and samples ($n = 5$). Color gradient represents gene expression as z-score.

cluster (Fig. 3c, d), a common MSC niche[39,40]. Known MSC
properties such as quiescence (G0) and the low gene regulation
dynamics[41–43] were determined by the cell cycle score and the
velocity of the scRNA-seq data, respectively. We identified a small
group of cells within the vascular myocyte cluster in the G1/G0
phase (Fig. 4a) using a computational assignment of cell-cycle
stage[44]. Cell velocity, which predicts the future state of individual
cells using the RNA splicing information from each cell[45], showed
that the same group of cells in G1/G0 phase in the vascular
myocyte cluster are depicted with low velocity vectors (Fig. 4b),
indicating low levels of transcriptional changes, another char-
acteristic of stem cells[46]. We defined cells within the vascular
myocyte cluster presenting with high expression of MSC markers
in a G1/G0 phase and with low velocity as the "MyoSPC" cluster.

**Integrating bulk SUSD2+/- RNA-seq and myometrial scRNA-
seq reveals a new MyoSPC marker (CRIP1).** Transcriptomic
analyses of SUSD2+/- bulk RNA and myometrial scRNA-seq

were performed, and the results were integrated to discover
possible overlapping MyoSPC markers. A total of 3700 DEGs
were found in the MyoSPC scRNA-seq cluster compared to the
rest of the myometrial cells (Supplementary Data 3). A little over
half (1929 DEGs) of the MyoSPC DEGs overlapped significantly
($p = 9.5 \times 10^{-81}$) with the DEGs from the SUSD2 +/- bulk RNA-
seq comparison (Fig. 5a and Supplementary Data 4). Correlation
analysis of the log$_2$ fold change (FC) in gene expression in the
scRNA-seq analysis with the SUSD2+/- bulk RNA-seq confirmed
that the MSC markers, *SUSD2*, *MCAM*, *PDGFRβ* and *CSPG4*
were upregulated in both (Fig. 5b). The most highly upregulated
gene in the MyoSPC cluster, Cysteine-Rich Intestinal Protein 1
(*CRIP1*), is also significantly upregulated in the SUSD2+ cells
(Fig. 5b). UMAP plot showed that *CRIP1* was highly expressed in
the vascular myocyte cluster (Fig. 5c), and more particularly in
the MyoSPC cluster (Fig. 5d). *CRIP1* expression wasn't differ-
entially expressed (log$_2$FC = −0.2, FDR $p = 9.9 \times 10^{-1}$) in the
RNA-seq results of the SP assay (Supplementary Fig. 2a).
Although the cell distribution in each cluster was different

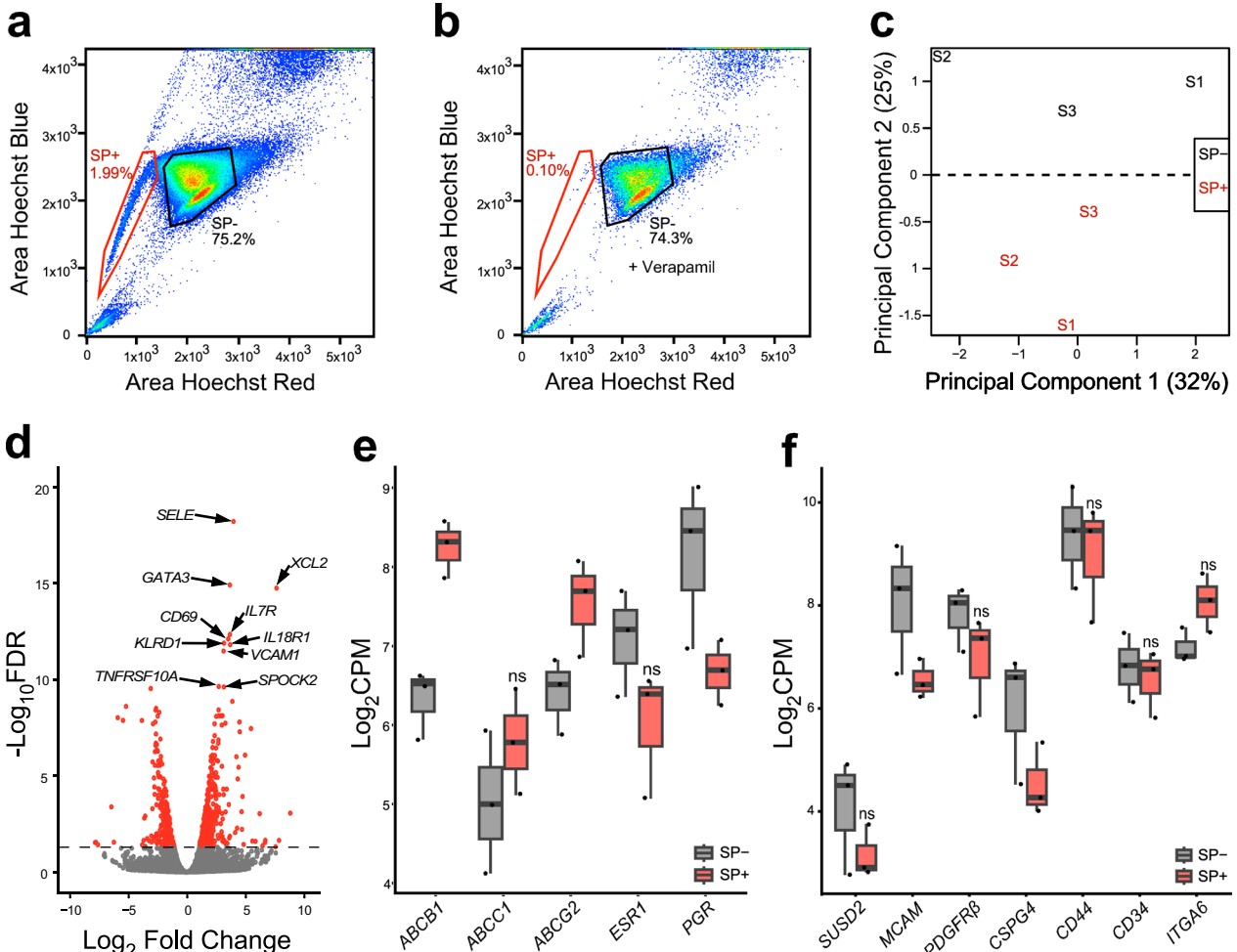

**Fig. 2 Transcriptomic analysis of the myometrium side population (SP). a** Scatter plot of the gating strategy to sort the SP+ and the SP- cells from human myometrium. **b** Verapamil pre-treatment of myometrial cells reduces the number of the SP+ cells from 1.99% to 0.1% of the total live single cells. **c** PCA plot of RNA-seq results from SP- (in black) and SP+ (in red) cells, each label represents one sample ($n = 3$), variance for each PC is indicated in percentage. **d** Volcano plot showing up ($n = 478$) and down ($n = 350$) DEGs with a false discovery rate (FDR) $p$-value < 0.05 in SP+ vs. SP- cells depicted as red dots, including *SELE*, *GATA3*, *XCL2*, *IL7R*, *CD69*, *KLRD1*, *IL18R1*, *VCAM1*, *TNFRSF10A*, and *SPOCK2*. Gray dots represent genes with an FDR $p$-value > 0.05. **e** Boxplots of myometrium SP associated genes, *ABCB1* ($\log_2 FC = 1.9$, FDR $p = 2.8 \times 10^{-5}$), *ABCC1* ($\log_2 FC = 0.7$, FDR $p = 5.9 \times 10^{-1}$), *ABCG2* ($\log_2 FC = 1.1$, FDR $p = 9.5 \times 10^{-2}$), *ESR1* ($\log_2 FC = -1.1$, FDR $p = 2.3 \times 10^{-1}$) and *PGR* ($\log_2 FC = -1.5$, FDR $p = 6.4 \times 10^{-2}$) and (**f**) MSC associated genes, *SUSD2* ($\log_2 FC = -1$, FDR $p = 5.6 \times 10^{-1}$), *MCAM* ($\log_2 FC = -1.5$, FDR $p = 3.9 \times 10^{-2}$), *PDGFRβ* ($\log_2 FC = -0.9$, FDR $p = 4.9 \times 10^{-1}$), *CSPG4* ($\log_2 FC = -1.5$, FDR $p = 3.9 \times 10^{-2}$), *CD44* ($\log_2 FC = -0.5$, FDR $p = 8.3 \times 10^{-1}$), *CD34* ($\log_2 FC = -0.2$, FDR $p = 4.9 \times 10^{-1}$), and *ITGA6* ($\log_2 FC = 0.9$, FDR $p = 2.1 \times 10^{-1}$) in the SP- (in gray) and SP+ (in red) cell population ($n = 3$), genes are expressed in $\log_2 CPM$. ns not significant, with FDR > 0.05.

(Supplementary Fig. 2b and Supplementary Data 5), we confirmed that *CRIP1* and the MSC markers were enriched in the MyoSPC cluster (Supplementary Fig. 2c) in cells from an orthogonal scRNA-seq study of myometrium from fibroid patients[38] when the cells were projected onto the UMAP shown in Fig. 3a.

**CRIP1+ cells have common stem/progenitor cell properties.** We next investigated if CRIP1+ cells presented stemness properties. Immunofluorescence analysis using 3D imaging of the myometrial layer showed that CRIP1+ cells are located surrounding the PECAM1+ vascular endothelial cells, a common MSC niche[39,40,47] (Fig. 6a and Supplementary Video). Interestingly, CRIP1+ cell immunofluorescence appeared to be predeominantly localized near the larger blood vessels and within a subset of SUSD2+ cells. Flow cytometry revealed that CRIP1+ cells represented between 2 to 5% of the total myometrial cells (Fig. 6b). PECAM1 was used for negative selection of the smaller

population of endothelial cells that also expressed CRIP1. CRIP1+/PECAM1- cells and the depleted cell population were sorted, and typical downstream stem cell assays were performed to determine if the CRIP1+/PECAM1- cells have stem/progenitor cell proprieties. Colony formation assays indicated that CRIP1+/PECAM1- sorted cells have a greater self-renewal capacity compared to the depleted sorted population (Fig. 6c), with a significant increase of 4.5-fold greater number of colonies formed (Fig. 6d), as well as a significant increase of the size of the colonies (Fig. 6e). After 5 days in smooth muscle differentiation media, CRIP1+/PECAM1- cells were positive for ACTA2, indicating that they differentiated into smooth muscle cells (Fig. 6f). Similarly, CRIP1+/PECAM1- cells were positive for Oil Red O staining (Fig. 6g), and alkaline phosphatase activity (Fig. 6h) when grown in either in adipogenic or osteogenic differentiation media, respectively, compared to CRIP1+/PECAM1- cells grown in control media, indicating that these putative MyoSPC cells have the capacity to differentiate into adipocytes and osteocytes.

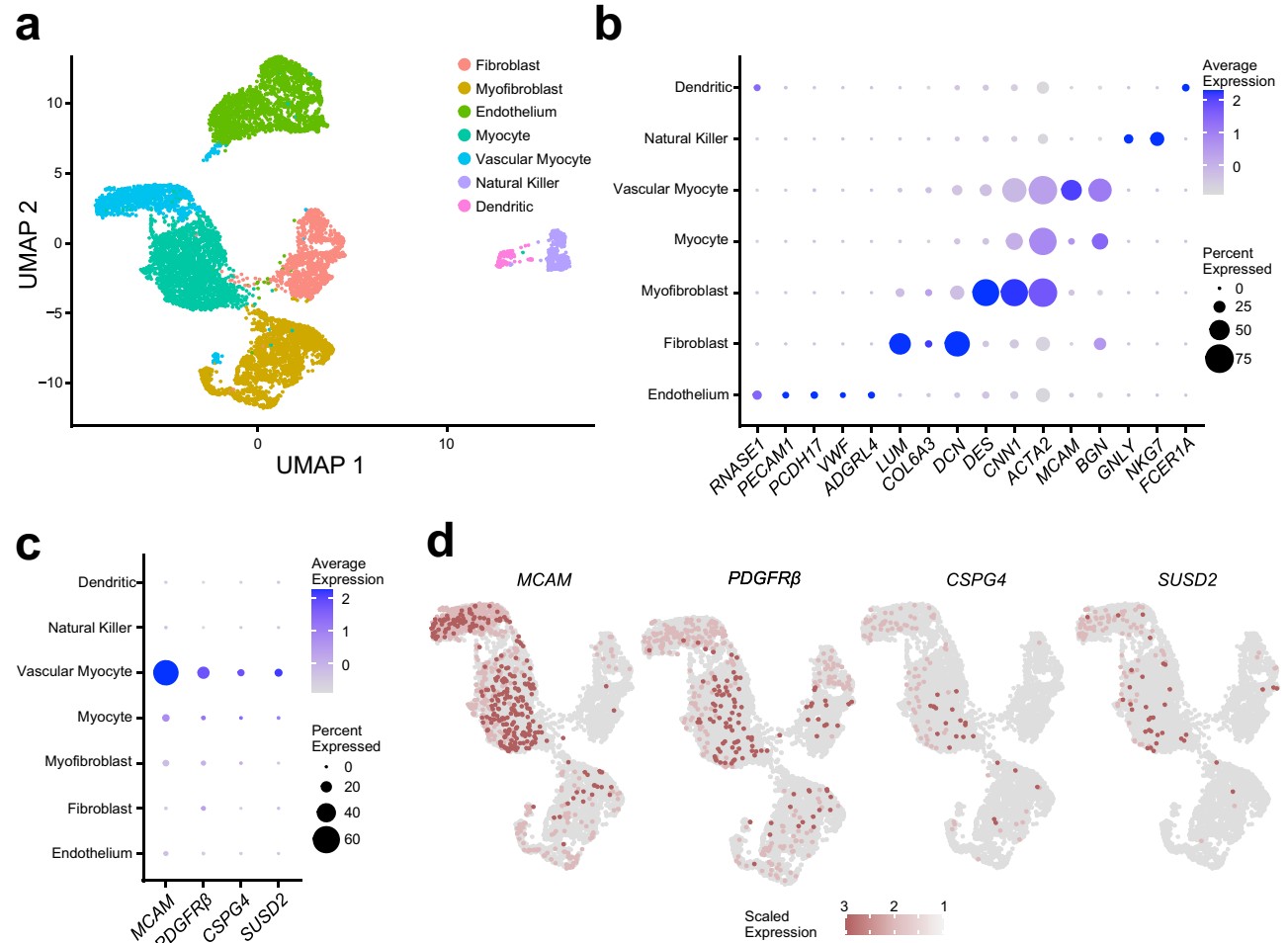

**Fig. 3 Single cell RNA-seq analysis of isolated cells from human myometrial samples. a** Uniform manifold approximation and projection (UMAP) visualization of 9775 isolated cells from human myometrial samples ($n = 5$). Each cluster ($n = 7$) represent a cell population with a similar transcriptomic profile. **b** Dotplot for cluster identification using specific markers for cell types found in the myometrium. MSC marker gene expression in the different myometrial cell clusters shown in a dotplot (**c**) and by UMAP (**d**). Average gene expression and percentage of cells expressing the specific gene in each cell cluster are shown by the color intensity and the diameter of the dot, respectively, in **b** and **c**. Color gradient in the UMAP represents gene expression as log$_2$CPM in **d**.

## Discussion

We have identified CRIP1 as a novel cell surface marker that enriches for a possible MyoSPC by combining the analyses of two next generation sequencing techniques, bulk RNA-seq from SUSD2+ and SUSD2- cells and scRNA-seq of total myometrial samples. Bulk RNA-seq, enriched for known MSC markers, but the large number of DEGs made it difficult to choose putative novel myometrium stem cell markers for further study. To reduce the number of candidate markers for validation and follow up studies, we used scRNA-seq to identify possible stem cells based on MSC markers. Subsequent stem cell assays confirmed that CRIP1+ cells have MSC properties, and further studies are underway to determine whether these cells could be a cell of origin for uterine fibroids.

CRIP1, Cysteine-rich intestinal protein 1, is a member of the LIM/double zinc-finger proteins that is predicted to be a novel biomarker in multiple cancers and can promote several biological processes, including cell migration, invasion and epithelial-mesenchymal transition by activating Wnt/β-catenin signaling, an essential pathway that maintains stem cell homeostasis in many tissues[48–50]. In an earlier study of SP+/- cells in fibroids[51], CRIP1 was among the DEGs detected by microarray analysis. In that study, the authors demonstrated

that the SP+ cells from fibroid tissues have stem cell characteristics, including self-renewal and differentiation into adipose and osteocyte cells. In the present study, we used myometrial samples from non-fibroid patients (MyoN) samples because we recently reported that myometria from fibroid patients (MyoF) have a different transcriptomic profile compared to MyoN samples, including an enrichment of DEGs in a leiomyoma disease ontology panel[52]. Here we have reported that CRIP1 expression was not differentially expressed and that the expression of the MSC markers, SUSD2, MCAM, PDGFRβ, and CSPG4 were decreased in SP+ compared to the SP- from MyoN cells. These discordant results could arise from the tissue type, that is, fibroid tumors versus MyoN, or because the SP technique could be more applicable to other tissues[20] or hematopoietic stem cells (HSCs). It is worth noting that HSCs exhibit a specific ABC transporter gene expression profile distinct from other stem cells, including MSCs[53]. HSCs expressed higher levels of most of the ABC transporters, including ABCB1, ABCC1 and ABCG2, compared to other stem cells. Indeed, the SP+ MyoN cells were enriched for ABC transporters and HSC-associated genes. Additionally, the SP technique relies on an intact cell metabolism and considerable variation in results has been observed[20].

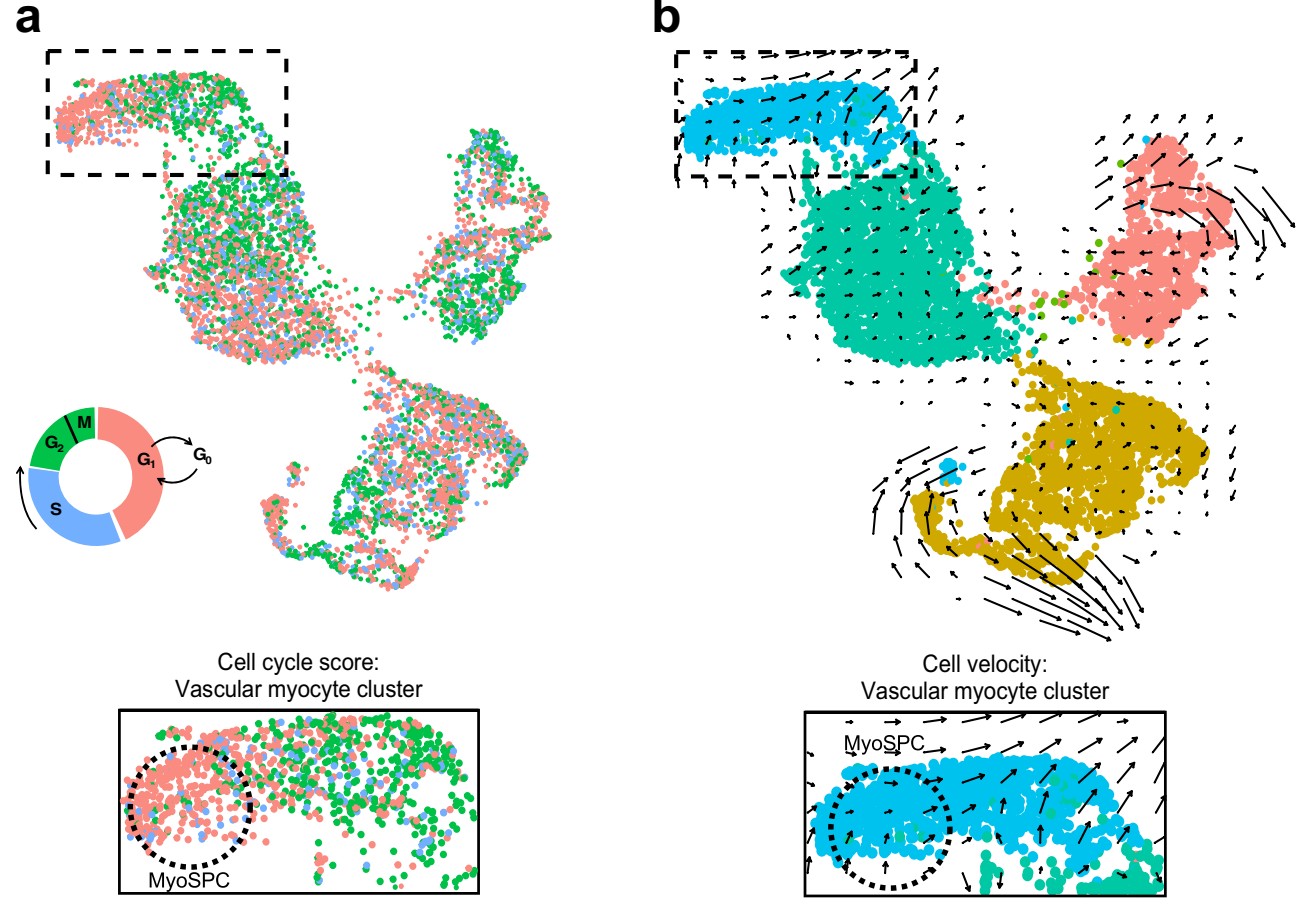

**Fig. 4 Identification of putative MyoSPCs from scRNA-seq. a** Cell cycle score for myometrial cells visualized in the UMAP plot. Cells in G1/G0, S, and G2/M phases are plotted with the corresponding color. Boxed area is shown at higher magnification. **b** Cell velocity predicting the future state of individual cells illustrated in a UMAP plotted with the clusters as in panel a showed that in the vascular myocyte cluster, the same group of cells in G1/G0 phase exhibit low velocity. Boxed area is shown at higher magnification. Putative MyoSPCs are encircled with black dotted lines.

Our scRNA-seq results suggested that human myometrium has at least 7 different cell types, including different types of smooth muscle cells, endothelial cells, and immune cells. Similar clusters were reported in a scRNA-seq comparison of fibroids and myometrium[38]. The depth of sequencing for each cell was close to saturation allowing us to identify a small cell population with stem cell characteristics, including the expression of MSC markers *SUSD2*, *MCAM*, *PDGFRβ*, and *CSPG4*, a quiescent (G0) cell cycle state[41], and low transcriptomic activity/low RNA velocity[41–43]. CRIP1+/PECAM1- cells were primarily located in the perivascular region, a common MSC niche[39,40,47], particularly by the larger myometrial blood vessels. These results were consistent with our scRNA-seq results showing that *CRIP1* expression was most highly expressed in the vascular myocyte cluster. Moreover, immunofluorescence staining showed that CRIP1+ cells were a subset the SUSD2+ cells and that CRIP1+/PECAM1- cells account for only 2 to 5% of the total human myometrial cells, a typical stem cell proportion in adult tissues[54]. Interestingly, the depleted population was able to form a few colonies, an indication that some cells in the depleted population also have self-renewal properties. Similar results were observed by us using other MSC markers to isolate myometrium stem/progenitor cells, including SUSD2, MCAM, or PDGFRβ[18]. This finding suggests that further enrichment of the MyoSPCs with some of the other MSC markers is possible or that myometrial cell plasticity is more common than previously appreciated.

Uterine fibroids are thought to be a clonal disease that we hypothesized to originate from a mutated MyoSPC. Inducing common mutations observed in fibroids in CRIP1+/PECAM1- myometrium cells, such as *MED12* (exon 2, 131 G > A, G44D) or overexpression of *HMGA2*, could provide insight into fibroid etiology and models to evaluate alternative therapeutics.

In summary, we have identified CRIP1 as a novel marker of MyoSPCs by integration of two transcriptome sequencing techniques, sorted bulk cell and single cell RNA-seq. Induction of a known fibroid subtype mutation in CRIP1 + /PECAM1- cells and their subsequent development into fibroid-like cells, could advance our understanding of fibroid etiology based on the hypothesis that a dysregulated MyoSPC is the origin of uterine fibroids.

**Study approval**. The use of human tissue specimens was approved by the Spectrum Health Systems and Michigan State University Institutional Review Boards (MSU IRB Study ID: STUDY00003101, SR IRB #2017-198) as secondary use of biobank materials.

## Methods
**Sample collection and cell isolation**. Myometrial samples from non-fibroid patients (MyoN) were obtained following total hysterectomy from pre-menopausal (aged 34-50), self-identified Caucasian women. No fibroids were detected by ultrasound prior to surgery. All patients who participated in the study gave consent

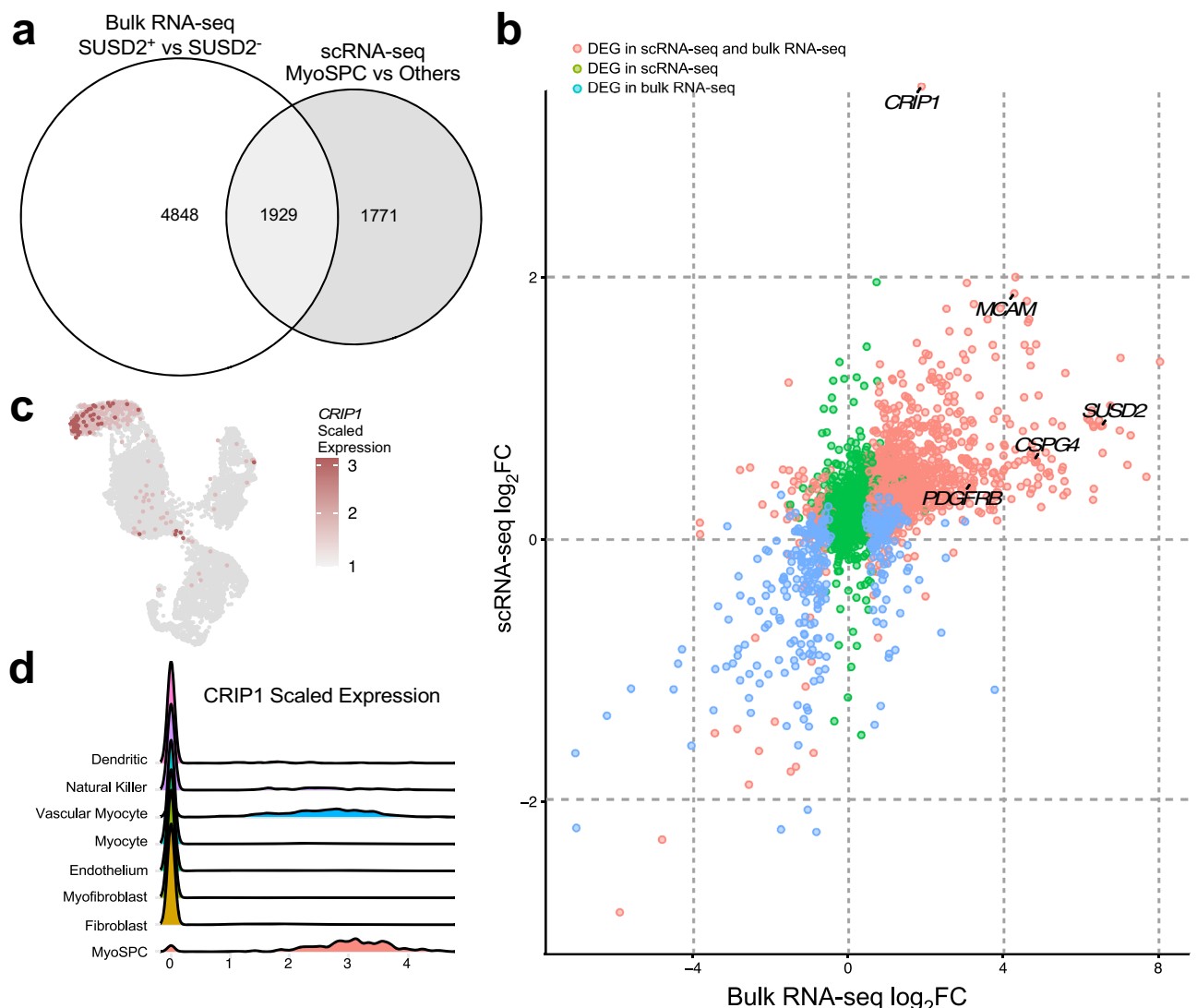

**Fig. 5 Integrated analysis of the bulk RNA-seq of SUSD2+ vs SUSD2- and the MyoSPC cluster vs the rest of the myometrial cells from the scRNA-seq.**
**a** Venn diagrams illustrate the overlapping DEGs between the bulk RNA-seq of SUSD2+ vs SUSD2- and the assigned MyoSPC cluster compared with the rest of the myometrial cells from the single cell RNA-seq analysis. **b** Scatter plot of Log$_2$ fold change genes from bulk RNA-seq (x-axis) and scRNA-seq (y-axis). DEGs in the bulk RNA-seq only were represented in blue dots, DEGs in the scRNA-seq only were represented in green dots, and the DEGs in both analyses were represented in red dots. *CRIP1* is highly upregulated in the MyoSPC cluster (log$_2$FC = 3.1, adjusted p-value = $4 \times 10^{-251}$) and in the SUSD2+ cells (log$_2$FC = 1.9, FDR = $5 \times 10^{-4}$). UMAP (**c**) and Ridge plots (**d**) display *CRIP1* scaled expression by cell cluster.

to donate tissue through the Spectrum Health Biorepository. Myometrial samples were washed with PBS, dissected away from non-myometrial tissue, and minced. Cells were isolated by incubation at 37 °C in baffled flasks containing digestion media (DMEM/F12, 10% fetal bovine serum (FBS), collagenase type I, DNAse type I, and MgCl$_2$) with agitation. The resulting cell suspensions were strained through 100- and 40-μm cell strainers, washed with warm media (DMEM/F12 containing 15% FBS), and centrifuged. Isolated cells were then stored in freeze media (90% FBS, 10% DMSO) at −80 °C until needed.

**Cell staining for FACS.** Human primary myometrial cells were thawed and resuspended in 1% bovine serum albumin blocking buffer for 20 min at room temperature (RT). Cells were then incubated with the primary antibody for 45 min at RT; SUSD2-PE anti-human (Miltenyi Biotec, #130-117-682), PECAM1-FITC anti-human (Thermo Fisher, #11-0319-42), and CRIP1 rabbit anti-human (Thermo Fisher, #PA5-24643). For CRIP1/PECAM1 staining, cells were incubated with an Alexa-647 anti-rabbit secondary antibody for 30 min at RT. Stained myometrial cells were then wash with flow buffer and resuspend in 1 mL of flow buffer with 1 μg of 4′,6-diamidino-2-phenylindole (DAPI) or Propidium Iodode (PI), depending on the experiment, for live-dead discrimination. Cells were sorted by the flow cytometry core at Van Andel Research Institute (VARI) using a FACSymphony S6 cytometer (BD Biosciences) and analyzed with FlowJo Software (BD Biosciences, version 10.8.1).

The side population assay was conducted as described previously[17]. Briefly, live cells were incubated with 5 μg/mL of Hoechst 33342 dye for 90 min. As a negative control, separate aliquots of cells from the same patients were treated with 25 μg/ml of verapamil (Sigma) prior to addition of the Hoechst dye. PI was added to stained cells with and without verapamil treatment and analyzed in a MoFlo Astrios (Beckman Coulter) for side population gating by Hoechst red and blue filters and sorting in media (DMEM/F12) at 4 °C. Examples of the gating strategy used for SUSD2$^{+/-}$, SP$^{+/-}$, and CRIP1$^{+}$/PECAM1$^{-}$ are shown in Supplementary Fig. 3.

**RNA isolation, library preparation and sequencing.** Total RNA was isolated from sorted cells using an RNeasy mini kit (Qiagen) and stored at −80 °C in nuclease-free water. RNA integrity values were determined with an Agilent 2100 Bioanalyzer (Thermo Fisher), and values ≥ 7.5 were used for library preparation and paired-end (2 × 100 bp) RNA-sequencing on an Illumina NextSeq 6000 instrument (Illumina). Libraries were prepared using a Kapa RNA HyperPrep kit with ribosomal reduction, pooled, and sequenced on flowcells to yield approximately 50–60 million reads/sample. Raw fastq files were deposited in the NCBI Gene Expression Omnibus (GSE234528).

For single cell RNA-seq, dead cells were removed from digested myometrial cells using the Dead Cell Removal Kit (Miltenyi Biotec, #130-090-101) per manufacturer's instructions. Live myometrial cells from 5 non-fibroid patients were then sequenced. Libraries were generated and sequenced using the 10X

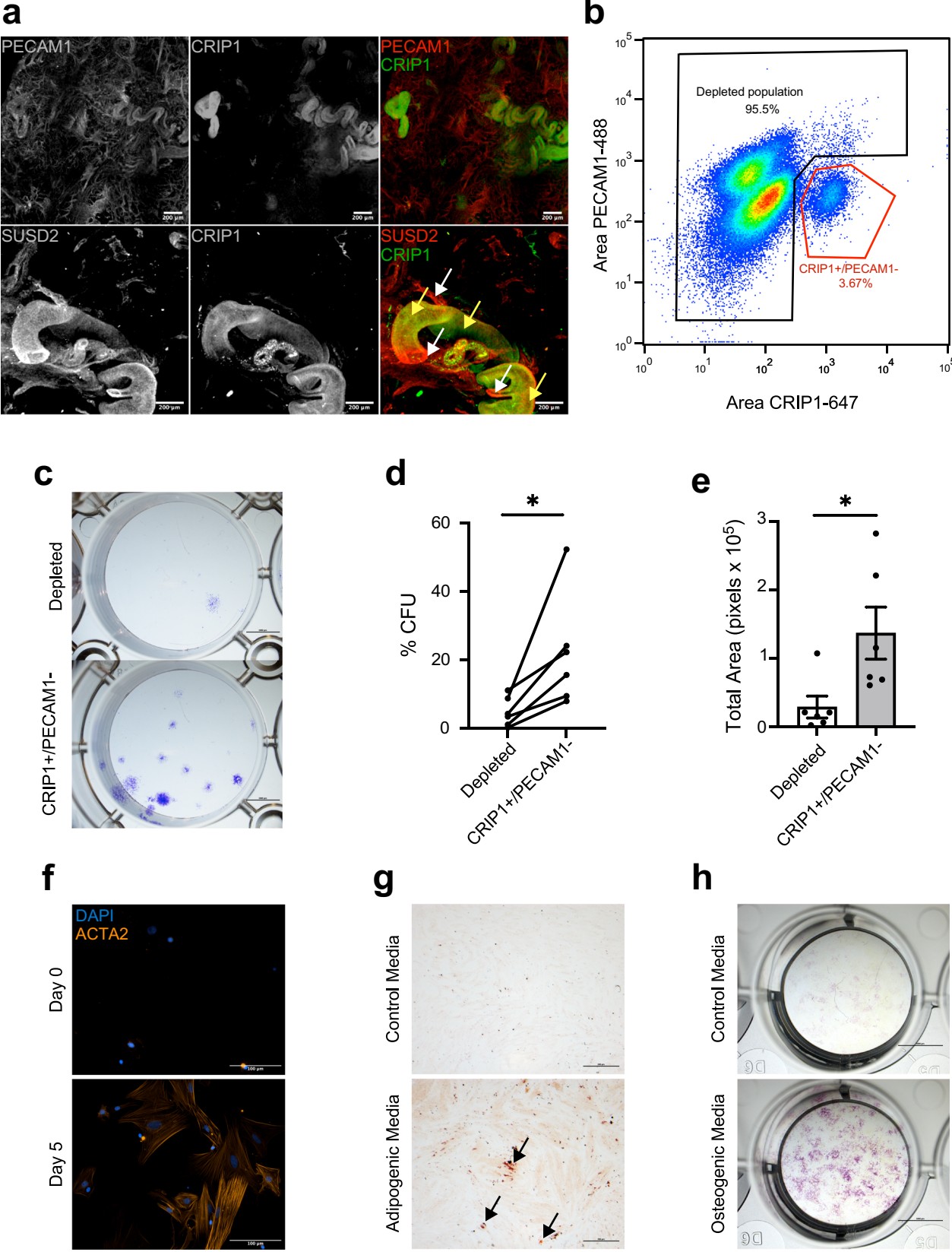

Chromium Next GEM Single Cell 3′ GEM kit (10X Genomics, v2) platform according to the manufacturer's instructions. 2 × 75 bp, paired end sequencing was performed on an Illumina NovaSeq 6000 sequencer using an S2 flow cell, 100 cycle sequencing kit (v1.5) to a minimum depth of 50 K reads per cell (Illumina Inc., San Diego, CA, USA). Base calling was done by Illumina RTA3 and output was demultiplexed and converted to FastQ format with Cell Ranger (10X Genomics, v3.1.0). Raw fastq files were deposited in the NCBI Gene Expression Omnibus (GSE234528).

**RNA seq analysis**. For bulk RNA-seq, reads were trimmed for quality and adapters using TrimGalore (version 0.6.5), and quality trimmed reads were

**Fig. 6 CRIP1+ cells have stem/progenitor cell characteristics. a** Representative ($n = 3$) immunofluorescence imaging of human myometrium using PECAM1 as an endothelial marker, SUSD2 as a mesenchymal stem cell marker, and CRIP1. Scale bar = 200 μm. **b** Representative ($n = 6$) scatter plot of the gating strategy for CRIP1+/PECAM- cell sort. **c** Representative ($n = 6$) images of colonies formed by the CRIP1+/PECAM- and depleted myometrial cells. **d** Plot of colony forming efficiency represented as %CFUs (#CFU/cells seeded × 100) of CRIP1+/PECAM- and depleted myometrial cells ($n = 6$). **e** Total area of colony formed in pixels from CRIP1+/PECAM- and depleted myometrial cells ($n = 6$). **f** ACTA2 immunofluorescence in CRIP1+/PECAM- myometrial cells after differentiation. Scale bar = 100 μm. Representative ($n = 3$) images of CRIP1+/PECAM- and depleted myometrial cells grown in control growth media and adipogenic (**g**) or osteogenic (**h**) differentiation media. Adipogenic and control cultures were stained with Oil Red O (red color, black arrows), and osteogenic and control cultures were stained for alkaline phosphatase activity (purple color). Scale bar for the adipogenic and osteogenic assays are 500 μm and 5 mm, respectively. *$p < 0.05$; by student $t$-test.

assessed with FastQC (version 0.11.7). Trimmed reads were mapped to Homo sapiens genome assembly GRCh38 (hg38) using STAR (version 2.7.9a). Reads overlapping Ensembl annotations (version 99) were quantified with STAR prior to model-based differential expression analysis using the edgeR-robust method with paired samples. Genes with low counts per million (CPM) were removed using the filterByExpr function from edgeR[55]. Scatterplots of two selected principal components was constructed with the PCAtools R package (version 2.5.13) to verify sample separation prior to statistical testing. Generalized linear models were used to determine if principal components were significantly associated with cell type. Genes were considered differentially expressed if their respective edgeR-robust FDR corrected $p$-values were less than 0.05. Differential expression was calculated by comparing SUSD2+ to SUSD2- cells and SP+ to SP-. DEGs were visualized with volcano plots and heatmaps generated using the EnhancedVolcano (version 1.8.0) and pheatmap (version 1.0.12) packages in R. Box plots of the $\log_2$(CPM) values were generated using the R package ggplot2 (version 3.4.0).

For scRNA-seq, demultiplexed sequencing reads were processed and aligned to the *Homo sapiens* genome assembly GRCh38 (hg38) using STAR (version 2.7.9a) with 10X Genomics Cell Ranger (version 3.1.0). Samples were merged using the integration anchors function of the Seurat package (version 4.2.1) from R[56]. Genes expressed in fewer than three cells in a sample were excluded, as well as cells that expressed fewer than 200 genes and mitochondrial gene content >5% of the total unique molecular identifier count. Data were normalized using a global-scaling normalization method[56] that normalizes the feature expression measurements for each cell by the total expression, multiplies this by a scale factor (10,000), and then log-transforms the results. The top 2000 most variable genes that were used for cell clustering were found using the *FindVariableFeatures* function and were then normalized using the *ScaleData* function. Based on an elbow plot generated using the *Elbowplot* function of Seurat, we selected 15 principal components (PC) for downstream analyses. Cell clusters were generated using *FindNeighbors* and *FindClusters* functions. For visualization, UMAPs were generated using the *RunUMAP, FeaturePlot* and *DimPlot* functions. The *DotPlot* Seurat function was used to generate dot plots to visualize gene expression for each assigned cluster. Cell cycle score and velocity were determined using the functions *CellCycleScoring* from Seurat and *RunVelocity* from SeuratWrappers (version 0.3.0), respectively. The "stem cell" cluster was selected using the *CellSelector* function from Seurat. The *RidgePlot* function from the Seurat R package was used for the visualization of *CRIP1* gene expression in the different myometrium clusters.

Venn diagrams of the overlapping DEGs from the bulk RNA-seq of SUSD2+ and SUSD2- cells and DEGs of scRNA-seq of the "MyoSPC" cluster compared to the rest of the myometrial cells were constructed using the eulerr package (version 6.1.1). The scatter plot of overlapping DEG from bulk- and scRNA-seq was generated with ggplot2 (version 3.4.0). *CRIP1* expression in the MyoSPC cluster was confirmed using a single cell data from the NCBI Gene Expression Omnibus (GSE162122). Single cell data from 5 MyoF from that study were mapped to the hg38 using STAR. A total of 18939 cells passed quality control and were projected onto our reference UMAP using the function MapQuery from Seurat.

**Imaging**. Whole mount immunofluorescent staining[57] was performed with human myometrial samples that were fixed in a 4:1 solution of methanol:DMSO. Tissue was removed from the fixative, rehydrated in (1:1) methanol: PBST (PBS, 1% triton) solution and washed in 100% PBST. Samples were incubated in a blocking solution (PBS, 2% powdered milk, 1% triton) and then stained with a 1:500 dilution for primary antibodies in blocking solution for 7 nights at 4 °C. Primary antibodies used were Rabbit anti-human CRIP1 (Thermo Fisher, #PA5-24643), Mouse anti-human SUSD2 (Biolegend #327401) and Mouse anti-human PECAM1 (Abcam, #ab9498). Samples were then incubated with secondary antibodies, Donkey anti-Rabbit IgG Alexa Fluor 555 (Invitrogen, #A31572), donkey anti-Mouse IgG Alexa Fluor 647 (Invitrogen, # A31571) at a dilution of 1:300 and Hoechst dye, for three nights at 4 °C. Samples were transferred to a methanol:PBST (1:1) solution, then washed in methanol and incubated at 4 °C overnight in a 3% $H_2O_2$ solution diluted in methanol. Tissues were then washed in methanol and cleared in benzyl alcohol:benzyl benzoate (1:2) overnight. Imaging was performed using a Leica SP8 TCS white light laser confocal microscope utilizing 10x air objective and a 7.0 μm Z stack[58]. Imaris v9.2.1 (Bitplane) commercial software was used to analyze confocal image files and create 3D renderings. The image files were imported into Imaris 3D

surpass mode and 3D renderings were created using the Surface plugin. Images were captured using the Snapshot plugin of Imaris and videos were generated using the Animation plugin.

**Colony formation and mesenchymal lineage differentiation**. CRIP1 + / PECAM1- and the depleted cell populations were sorted as described above and plated at 50 cells/cm$^2$ in triplicate in growth media (DMEM/F12, 10% FBS) overnight, then grown in MesenPro RS (Thermo Fisher, # 12746012) for 2 to 3 weeks. Cultures were fixed in 4% paraformaldehyde (PFA) and stained with crystal violet for colony visualization. Colonies were counted and total surface area was estimated using ImageJ (version 1.53k), and the percent colony-forming units (CFUs) was calculated as (number of colonies/number of cells plated) × 100 and averaged for triplicates. Matched CRIP1 + /PECAM1- and depleted populations were cultured in different wells of the same plate, and both cell types were assayed on the same day. Images were taken using a Nikon SMZ18 microscope and Ds-Ri1 camera (Nikon Instruments Inc.). For osteogenic and adipogenic differentiation, CRIP1 + /PECAM1- cells were plated in triplicate at 50 to 80% confluency in growth media (DMEM/F12, 10% FBS) overnight and then cultured for 10 days in fresh StemPro Adipogenesis Differentiation (Thermo Fisher Scientific, #A1007001) or StemPro Osteogenesis Differentiation (Thermo Fisher Scientific, #A1007201) media according to the manufacturer's instructions. Cells were cultured in regular growth media to serve as differentiation controls. To assay adipogenic differentiation, cultures were fixed in 4% PFA and stained using Oil Red O (Sigma, #01391) according to the manufacturer's instructions. To assay osteogenic differentiation, cultures were stained for alkaline phosphatase activity using the Alkaline Phosphatase (AP), Leukocyte kit (Sigma, #86R-1KT) according to the manufacturer's instructions. For smooth muscle differentiation, cells were plated on 1 mg/ml dried rat tail collagen (Corning, #354 236) in 8 well chamber slides with growth media (DMEM/F12, 10% FBS) overnight and then cultured in Medium 231 with a smooth muscle differentiation supplement (Thermo Fisher, #M231500). Cells were fixed in 4% PFA at the indicated times (D0: before adding the differentiation media, D5: 5 days in differentiation media) and stained using αSMA-Cy3 (Sigma, #C6198). Images were taken using a Nikon Eclipse Ni-U or Nikon SMZ18 microscope and Ds-Qi1MC or Ds-Ri1 camera (Nikon Instruments Inc.).

**Statistical and reproducibility**. Bioinformatic statistics were performed using the listed packages in R (version 4.0.2). DEGs of the bulk RNA-seq from SUSD2+ vs. SUSD2- and the SP+ vs. SP- were identified as those having an Benjamini–Hochberg FDR corrected $p < 0.05$[59]. Data with unequal variances were log transformed, and homogeneity of variances verified before completion of analyses. DEGs of the scRNA-seq were calculated using the non-parametric Wilcoxon Rank Sum test. Adjusted $p$-value, based on Bonferroni correction using all features in the dataset was used to determine significance. DEGs with adjusted $p$-value < 0.05 were consider as significant. Hypergeometric testing was performed using the function phyper from R. For the colony formation assays, comparison of two means was performed with a two-sided student $t$-test, and significance was determined at $p < 0.05$ after confirming normal distribution using Graphpad Prism (version 9.4.1).

**Reporting summary**. Further information on research design is available in the Nature Portfolio Reporting Summary linked to this article.

## Data availability

Raw fastq files were deposited and are freely available in the NCBI Gene Expression Omnibus (GSE234528). Data used in Fig. 6d, e are available in Supplementary Data 6 and 7, respectively. All other data are available from the corresponding author upon reasonable request.

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

## Acknowledgements

We would like to thank the patients who consented for the study, the Spectrum Health Systems Universal Biorepository staff, and the Van Andel Institute Genomics Core (RRID:SCR_022913), especially Marie Adams, Rebecca Siwicki and Julie Koeman, for their assistance with the construction and sequencing of 10X libraries for the single cell RNA-seq and bulk RNA-seq. This work was supported by grants R01HD096259 and R21HD100959 from the Eunice Kennedy Shriver National Institute of Child Health and Human Development (NICHD) to J.M.T, March of Dimes (5-FY20-209) and NICHD R01HD109152 to R.A. and the SRI/Bayer Discovery/Innovation Grant (to E.N.P).

## Author contributions

Experimental design (E.N.P., J.M.T.) collected data and performed experiments (E.N.P., T.J.C., S.F.), analyzed data (E.N.P., R.S., K.H.L., R.A., J.M.T.), wrote/reviewed manuscript (E.N.P., T.J.C., S.F., R.S., K.H.L., R.A., J.M.T.).

## Competing interests

The authors declare no competing interests.
