## [Peer Review File · Communications Biology]

REVIEWERS' COMMENTS:

Reviewer #1 (Remarks to the Author):

Comment on Paul et al. Paper # COMMSBIO-23-1323-T

Uterine fibroid is a multifactorial disease and many questions remain unanswered. Available evidences suggest that myometrial stem/progenitor cells may have an important role in fibroid development. However, due to complexity of myometrium specific stem cell markers, the role of stem cells in studying fibroid pathobiology has been challenging. This study introduced a novel marker CRIP1 for myometrial stem/progenitor cells (MyoSPCs). This manuscript is well written and findings are profoundly discussed.

The following specific comments are provided.

Specific comments:

- (1) Originality: This work is original and can potentially have impact on fibroid research.
- (2) Did authors quantify the levels of stemness markers like KLF4, NANOG, SOX2, and OCT4 in CRIP1-positive cells? A brief comment on this would be welcomed.
- (3) The stemness of CRIP1-positive cells was demonstrated by their in vitro colony-formation capacity and mesenchymal lineage differentiation. However, the in vivo tumor-regeneration ability of CRIP1-positive cells should be studied.

Reviewer #2 (Remarks to the Author):

Uterine fibroids, or leiomyomata, are seriously under-researched, although they affect a large number of women and, by implication, families. The origin of these benign tumours is still unclear, with dysregulated stem cells one of the major hypotheses. The manuscript by Paul et al. proposes a new marker for myometrial stem/progenitor cells (MyoSPCs) in Cysteine-Rich Intestinal Protein 1 (CRIP1), which the authors identified using a clever combination of single cell RNA-sequencing and bulk sequencing on populations enriched for known SPC markers.

This new marker, CRIP1, seems to indeed further enrich the SPC compartment within the myometrial cells investigated here. Questions of low n-numbers and sample size aside (as they always arise in sequencing debates), I think this is an exiting finding that should be shared with the leiomyoma-field without delay.

A few notes:

BABB seems a dangerous method to use in tissue clearing by now, as there are better alternatives available now, e.g., CUBIC or X-CLARITY. I hope the authors will have the opportunity to switch to one of those less toxic methods in the future.

The language is somewhat precious at times, e.g., line 176: 'We next investigated the CRIP1+ cells to establish their stem cell bona fides.' - better: 'to validate/confirm their stem cell character', 'properties' ... 'bona fide' is used as an adverb.

Also line 248: 'heretofore' - just say 'previously', as there might be few lawyers among the readership of Comms Bio.

Could the authors speculate about the relation of the CRIP1+ cells and the known mutations (MED12/HMGA2/FH/COL4/6) found in leiomyomata, please?

Answer to Reviewers:

Reviewer #1:

(1) Originality: This work is original and can potentially have impact on fibroid research.

Thank you for reviewing the manuscript and for the comments.

(2) Did authors quantify the levels of stemness markers like *KLF4*, *NANOG*, *SOX2*, and *OCT4* in CRIP1-positive cells? A brief comment on this would be welcomed.

We appreciate the comment, unfortunately the depth of sequencing in the single cell RNA-seq didn't allow us to detect of *SOX2* and *OCT4*. As shown in the UMAP below, *NANOG* was expressed by few cells and *KLF4* was not specific at the RNA level.

(3) The stemness of CRIP1-positive cells was demonstrated by their *in vitro* colony-formation capacity and mesenchymal lineage differentiation. However, the *in vivo* tumor-regeneration ability of CRIP1-positive cells should be studied.

In vivo experiments, such as the kidney capsule xenotransplantation using mutated CRIP1+ cells will be performed in future studies.

Reviewer #2 (Remarks to the Author):

BABB seems a dangerous method to use in tissue clearing by now, as there are better alternatives available now, e.g., CUBIC or X-CLARITY. I hope the authors will have the opportunity to switch to one of those less toxic methods in the future.

Thanks to review the manuscript and we appreciate the recommendation on the tissue clearing.

The language is somewhat precious at times, e.g., line 176: 'We next investigated the CRIP1+ cells to establish their stem cell bona fides.' - better: 'to validate/confirm their stem cell character', 'properties' ... 'bona fide' is used as an adverb.

Also line 248: 'heretofore' - just say 'previously', as there might be few lawyers among the readership of Comms Bio.

Thanks for the suggestion, we modified the two sentences accordingly.

Line 171: "*We next investigated if CRIP1+ cells presented stemness properties.*"

Line 245: "*This finding suggests that further enrichment of the MyoSPCs with some of the other MSC markers is possible or that myometrial cell plasticity is more common than previously appreciated.*"

Could the authors speculate about the relation of the CRIP1+ cells and the known mutations (MED12/HMGA2/FH/COL4/6) found in leiomyomata, please?

We added the following sentence in the discussion.

Line 248: "*Uterine fibroids are thought to be a clonal disease that we hypothesized to originate from a mutated MyoSPC. Inducing common mutations observed in fibroids in CRIP1+/PECAM1- myometrium cells, such as MED12 (exon 2, 131G>A, G44D) or HMGA2 overexpression, could provide insight into fibroid etiology and models to evaluate alternative therapeutics.*"